# Critical Environmental Education in Latin America from a Socio-Environmental Perspective: Identity, Territory, and Social Innovation

Rodrigo Florencio da Silva [1,*], Alma Delia Torres-Rivera [2], Vilmar Alves Pereira [3], Luciano Regis Cardoso [4] and Melgris José Becerra [5]

1 ESIME Ticomán del Instituto Politécnico Nacional, Mexico City 06600, Mexico
2 UPIEM del Instituto Politécnico Nacional, Mexico City 07738, Mexico; atorresri@ipn.mx
3 School of Education, Universidad Internacional Iberoamericana—UNINI, Campeche 24560, Mexico; vilmar1972@gmail.com
4 Instituto de Desenvolvimento Sustentável Mamirauá, Tefe 69553-225, Brazil; luciano.rcardoso@gmail.com
5 Instituto de Geociências, Universidade Federal do Pará (UFPA), Belem 66075-110, Brazil; jose.becerra.ruiz@gmail.com
* Correspondence: rflorencio@ipn.mx

**Abstract:** The objective of this study was to contemplate the role of critical environmental education in Latin America from a socio-environmental perspective and explore how environmental problems associated with justice in territories and communities face the dynamics of the complexity of the effects of climate change. They modify the economic and social dynamics that little by little strip communities of their identity and deepen inequality. Selection and recovery of the articles in the bibliographic review, published between 2018 and 2022, used to determine the state of the question were carried out with the search chain integrated by the following keywords: critical environmental education, territory, and social innovation, which make up an analysis carried out using hermeneutic phenomenology from a socio-environmental perspective. The main finding is that critical environmental education in Latin America reveals historical distortions, forms of colonization, and modes of production associated with the exploitation of nature that deepen extreme poverty. On the other hand, the region's contradictions contribute to understanding the territory and identifying processes of social innovation that favor community life, recognizing new ways of being and living together in Latin America, whose cultural dimension and belonging follow the logic governed by the principles of binomial biodiversity and economy.

**Keywords:** critical environmental education; Latin America; identity; territory; social innovation

## 1. Introduction

Explicating the changes caused by the environmental problems experienced in 21st-century society demands understanding the interdependence between communities and ecosystems, which shapes their interactions in territorial frames [1–4]. The exchanges offer one way of approximating the socio-environmental problems that relate to identity [5], territory [6,7], and social innovation [8] by articulating social structures. In this approach, critical environmental education (CEE) is a mechanism for attaining inter- and intra-generational environmental justice in Latin America. However, the theoretical approach that dominates ecological studies aligns with the Western perspective, ignoring the indirect or tangential impact on territories and communities. Overcoming this fragmentation requires integrating socioeconomic and political change patterns associated with exploiting natural resources and legitimized by progress, modernization, and globalization in the region's territories [7].

Historically, progress in society was conceived as a strategy for solving the effects caused by the Second World War [9]. Economic models focused on capital accumulation [10] but often disregarded the environmental and social impacts experienced by the affected societies. The intensive use of natural resources and economic practices focused on the accumulation of capital by a few and left most of the population in poverty [11], which energized the emergence of popular environmentalist movements (MPAs) in the 1970s. With some advances and setbacks, MPAs have fueled resistance and struggles for emancipation in the face of limited democratic guarantees and increasing threats to living conditions.

Here, individuals in the dynamics of a collective action face environmental difficulties by innovating with/from/toward their communities [12]. This set of inter-relationships is a factor in the social transformation of socio-environmental problems that assigns a crucial role to CEE as a means of strengthening solidarity by driving community dynamics consistent with the identifying characteristics of their territories.

This study recognizes the plurality, diversity, and richness of worldviews about Latin America. It takes as a reference the understanding of Latin America as one of the most biodiverse regions in the world, marked by its cultural richness and multiple potentialities. However, paradoxically, it is one region with the most significant socio-environmental problems: social inequality, extreme poverty, and hunger.

Based on this, the study highlights the importance of CEE in Latin America, valuing the region's cultural and environmental diversity. Furthermore, the need to overcome the capitalist and anthropocentric development model is emphasized, promoting approaches that integrate ecological conservation and social justice. Additionally, the significance of recognizing and valuing the region's knowledge and philosophies, challenging Eurocentric thinking, is emphasized. Collectively, these contributions aim to foster a transformative approach to environmental education that promotes a more sustainable and equitable future for Latin America. Finally, it deals with the need for social innovation perspectives as guiding principles for new forms of social intervention oriented toward life sustainability with an analytical approach.

The structure of the work, the axial axis of CEE, is the integration of social, political, and economic dimensions in the learning process so that students critically analyze environmental problems, considering the dynamics of power, social inequalities, and historical contexts. CEE seeks to foster a deep understanding of the complex inter-relationships between humans and the environment while promoting social innovation with an orientation toward environmental justice and community participation within the sustainability framework.

This study is organized through a literature review encompassing topics such as CEE, identity, territory, and social innovation from a socio-environmental perspective. The methods employed include hermeneutic phenomenology, and the results are discussed along with additional reflections.

## 2. Review of the Literature

The literature review was organized logically and coherently, following a thematic approach. Various aspects of the main topic were analyzed, such as critical environmental education, identity, territory, and social innovation from a socio-environmental perspective. Each theme was explored by reviewing and analyzing relevant studies, theories, and approaches in the field. As a result, connections were established, and debates and gaps in the existing literature were identified. The organization of the literature review allowed for a comprehensive view of the topic and provided a solid foundation for further study development.

### 2.1. Critical Environmental Education and Identity

The socio-environmental issues in Latin America stem from social inequality, as the unequal distribution of income significantly deviates from the goal of democratizing the utilization of natural resources. This discrepancy arises due to extensive extractive practices and the lack of effective policies within the existing development model [13]. Consequently,

addressing this problem necessitates two crucial steps: (a) a comprehensive reassessment of historical colonialist processes and (b) an exploration of alternative ethical practices that foster new ways of coexistence [14].

The link between environmental education and social criticism is an essential aspect explored in this study. Educational institutions play a crucial role in promoting awareness of the environmental impact of human actions and advocating for equitable resource distribution for environmental protection [15]. CEE is identified as a pedagogical approach that not only imparts knowledge about socio-environmental issues but also takes proactive measures to address the injustices caused by the prevailing culture of natural resource exploitation [16,17]. By integrating social criticism into environmental education, CEE seeks to foster a deeper understanding of the interconnectedness between social and environmental issues, aiming to promote sustainable practices and a more equitable future.

Environmental education must link ecological and social processes in the world by intervening and existing in nature [18]. The environmental and social problems faced by communities since the 1950s and 1960s have created platforms for reflection and critical awareness regarding the industrialization schemes established in Latin America. The contribution of CEE is to establish platforms for debates to develop essential social actors that promote and understand socio-environmental problems in their multiple dimensions [19], which connect nature and society through reflection and enable the praxis of socio-environmental transformation in the contemporary world.

In Latin America, identity is a dense movement that is relied upon as a way of defining and locating oneself in society [20,21], which links people in social groups [22] by finding a cognitive motivation and emotional coincidence appropriate to their time and history and regulates behaviors in various domains [23,24]. Furthermore, people can define themselves in different roles by coexisting in multiple communities, regardless of the environmental dimension [25]. Therefore, identity motivates people to feel, perceive, and behave in other contexts [20].

It is essential to recognize the aspects that directly interfere with this identity to demarcate it. In this sense, some problems experienced have already been identified in the structure of the colonization processes, generating greater cultural, social, and economic dependence. One example of how it manifests itself can be identified today on the horizon of neoliberal policies imposed on the region with their directions [26]. To achieve this, it is necessary to (a) understand the social equality of individuals; (b) strengthen identities as sources of social equality of individuals; (c) recognize that communities and their membership, as well as their identity, are the product of free and autonomous decisions of free and autonomous individuals; (d) understand the reciprocity between socially equal individuals; (e) equally redistribute resources and products; and (f) study the association of the population on a global, local, and regional scale.

The model of epistemological decolonization refers to the cultures and ontological propositions of other ways of living and inhabiting the world. Latin America and the "indigenous" population occupy a basal foundational position in the constitution and history of the coloniality of power [26]. Hence, their current place and role in the epistemic/theoretical/historical/aesthetic/political subversion of this energy pattern in crisis were implied in the proposals of the global decoloniality of power and for a good life as an alternative social existence [27].

To face these challenges, CEE calls for redoubled learning and confronts a modus operandi already rooted in Latin American culture that continues to perpetuate. Simultaneously, it is the first call for confrontation on violence against indigenous people and local communities, whose existence is threatened daily in defense of their territories [28,29].

Resistance movements to the violence suffered by people and communities claim broader visions of identity and a search for alternatives through social innovation [30]. Thus, the horizon of CEE also points to possible forms of social intervention through CEE. Hence, overcoming social inequality and environmental degradation [31] are key tasks for CEE and social innovation in its praxis.

As previously mentioned, these broad CEE challenges do not occur in empty spaces but, on the contrary, are drenched in multiple contexts in which life is mitigated daily. That is why we must recognize the potential of CEE from the territories where, as of now, there is the existence of socio-environmental conflicts, as well as the conditions for the construction of other possible forms and modes of resistance and social organization and permanent dialogue from the perspective of social innovation.

### 2.2. Territory and Socio-Spatial Development in Liberation Praxis

Historically, the social inequality and environmental degradation experienced in Latin America demand critical reflection [32] in search of a practice to overcome them. To search for a liberating praxis, a constant movement of theory and practice is required toward the following triad: territory, socio-spatial development, and liberation. These polysemic concepts are assumed banners of struggle or axes in constructing sustainable environmental and social justice. The antecedents of inequality and environmental degradation are founded on the dualism between society and nature, mediated by Western culture [33]. However, other possibilities for the relationship between humanity and nature integrate human rationality from culture and history [34].

Human relations have their material basis in culture in a broad, heterogeneous, and situated manner in time, space, territorial constructs, and their interrelations with nature. Thus, human relations are mainly based on culture, which considers all cultural manifestations heterogeneously with a specific spatial and temporal scale based on territory and its relationship with nature [35].

Social groups in the geographical area possess material and symbolic relations in the form of complex interactions [36] from the practice of a controversial geographical concept. Still, they are widely assimilated in the political agenda of social movements in Latin America: the "territory". Distributed across the physical terrain of Latin America, many complex and diverse relationships between social groups and their contexts coexist. This diversity of ways of living, occupying the physical and subjective concretion of geographical space concerning conflict and cooperation that generate disputes and confluences of power, shapes the concept of the territory.

The territory can be conceptualized as a fluid border or "force field", simultaneously with space and social relations [37]. The region is the power exercised in social relations projected on a "material, spatial substrate of reference". Therefore, it is a socio-spatial concept [37]. In this view, territories reflect historical dynamics [38] and are not on a long-term scale. Thus, spatial expressions of the "domination of nature" of the capitalist mode of production coexist with specific social groups that maintain traditional social relations and precapitalist forms of economy that embody less aggressive ways of life [37].

From this socio-spatial concept, overcoming social inequality and environmental degradation must delineate the praxis for CEE with one proposal that must act from diverse edges. The most usual is the adoption of neoliberal policies that argue that development is synonymous with economic growth. Contrarily, an adequate response is that economic development, in the best interest, can strictly be only a means and never an end [39]. Specifically, considering that sustainability is quite far from the plan of economic development in Latin American countries because of strategies that continue to perpetuate the ethical and political ideologies of a capitalist model, a decision favoring a debate on natural resources and landscape heritage linked to the territories is still pending [39].

Economic development must address the challenges faced by society, because the effect of environmental degradation is more severe than the flaws of the markets. However, a holistic solution cannot emerge through adjustments [31], because explosive social exclusion owns the capitalist periphery.

To endorse this other perception of development as a historical construction of an endogenous impetus, the concept of social innovation has much to contribute. Social innovation is recognized as a response that targets and generates social change and simultaneously links three characteristics [40]. First, innovation is related to an initiative to escape

the established order. Second, it is a new way of thinking or doing something, a qualitative social change, an alternative, or even a break from traditional processes. Third, the most common origin of social innovation is society's autonomy.

Autonomy is defined as the conscious self-institution of society based on the political guarantee and the practical material possibility of equal opportunities for participation in decision making to open a radically alternative field [39]. Thus, the autonomy of a group to adopt a specific concept of development, or more broadly a particular way of life, requires the consideration of this group not in isolation but in the context of its relationships with other groups in its territory.

Social innovation should be present in public policies [40,41]. However, understanding the root causes of environmental problems in territorial frames and the community in the global eco-economic system needs to catch up on social innovation. In any case, mobilization around environmental justice [42] generates a collective action informally by a widespread movement [43]. The popular environmental movement in the Latin region is an organized pronouncement to act. Ecological problems and CEE have challenged the social structures and processes that shape the development of people and territories [40].

When considering social innovation from socio-spatial development as a process of overcoming social inequality and environmental degradation [44], it is vital to mention its most famous proponent, education. However, it is not just any education that possesses the capacity to promote social innovation that we defend here. It must be a critical, dialogic, and contextualized education to transform real situations of oppression and degradation. From a critical–environmental paradigm, CEE contributes to constructing alternatives for new ways of relating and living together based on ethical possibilities. Therefore, it is a fundamental strategy for socio-spatial development and social innovation [45] to overcome the challenges faced by the people of Latin America.

### 2.3. Social Innovation from a Socio-Environmental Perspective

The concept of social innovation is valuable for analyzing socio-spatial development and environmental education [46]. First, however, it is necessary to review different perspectives of its approach. Specifically, it argues that addressing both environmental education and territorial development in the process of social innovation [47] are critical components of social innovation in the context of Latin America, which is intrinsic to the identity of its communities [46].

Social innovation must seek more equity between citizens and social groups through market mechanisms that are socializing [48], which creates the necessary conditions for opportunities to make the market more inclusive and strengthen the supportive content of social relationships between people involved in social innovation initiatives, as well as invoke these relationships as triggers of socio-political empowerment.

Social innovation is classified as (1) a concept significant in business administration, public debate, scientific research, and ethical controversy [49]; (2) collective actions and social relationships that address social problems neglected by the public sector or the market [50]; and (3) something that can be "done", as if social innovation becomes a "thing", a process that in the best case can be separated from its context [51].

The importance of changes in the structure of territories in the function of markets confirms disruptive processes in the social sphere, as they create ways for social innovation from communities [52]. Hence, an exploration of the sequential deployment of innovation processes to democratize structures is to become a critical axis of sustainable development, and CEE is the key to the transformation of collective action because of its response to the socio-environmental needs of society at the dawn of the 21st century.

Social innovation is fundamentally about achieving territorial development. However, innovation tends to break free from the impositions of organizational structures by acting against them to synthesize new social cohesion and collective action toward a sustainable society. Social innovation is usually associated with territory [43]. Environmental problems, through reflection and action, take on a relevant significance from changes in the economic,

political, and cultural order present in the construction of significant new opportunities as an answer to challenge social inequality.

The relationship between society and territory catalyzes community development initiatives based on the inclusion of excluded groups [53]. Thus, the central axis is social inclusion for obtaining solutions to the problems of a society in territories. In addition, justice is inherently a social innovation from the perspective of ethical positions, environmental problem choices [54], and social values to intentionally plan and implement alternatives to solve inequality and ecological degradation problems [55].

Social innovation introduces tension between individual and collective actions elaborated when trait market dynamics for territorial development are implemented [46]. Individuals want to overcome their conditions of exclusion. Social structures cannot simply be creative forces of collective actions that internalize norms, customs, and identities that shape inequalities; even if the knowledge is available, this explains the working of natural and social worlds and their interactions with CEE.

Here, the paradox of social innovation is the process of disruptive changes in the social order to internalize the dynamics and consequent resurgence of a territorial development policy. It is evident that, in the long run, social innovation as a joint function will end up questioning organizational structures, as the social and solidarity economy did in the 19th century. We propose thinking about social innovation within the framework of territorial development; it would not exhibit the capability or incapability of execution but establish the role of CEE. Environmental education is part of the evolutionary process of the human species that endows it with the capacity to think about new ecologies and economic policies for social innovation. Based on practical awareness and emancipatory interest, CEE is a vital axis of freedom from an ideological (and material) perspective. It also offers tools to assess all the options available to realize the full potential for being involved in changing structures and processes of social innovation through individual and collective action.

Two types of social innovations have been proposed: pure and desirable [56]. Pure social innovations are not entrepreneurial innovations and have no business potential. Instead, they target needs that still need to be satisfied by market innovations. Simultaneously, "desirable" social innovations expand the set of reasonable options from a normative point of view. Furthermore, pure social innovations have the character of public goods; hence, they are often the responsibility of welfare economics. Therefore, to expose the scope of social innovation, the literature often tries to offer strategies and roadmaps for creating social innovations rather than explaining social innovation within a territorial context.

Social innovations transform territories, and this transformation exceeds the added value generated by "innovative activities and services that are motivated by the objective of satisfying a social need and are predominantly developed and disseminated through organizations whose primary purposes are social innovations" [57]. Thus, social innovation and territorial development form a reciprocal relationship; one does not exist without the other. Unfortunately, however, recognizing this reciprocal relationship leads to counterproductive effects.

To analyze the boundaries between social innovation and technology, it is necessary to understand their history and what drives or impedes their evolution. Often, it must be understood that the technique represents a realization of changes in organizational structures influenced by technical inventions characterized by their simplicity, low cost, ease of application, and guaranteed social impact in solving social problems.

Social innovation is "a novel solution to a social problem that is more effective, efficient, and sustainable than existing solutions, and for which the value created accrues primarily to society as a whole rather than individuals" [58]. Innovation is social only if the value created prioritizes social value generation rather than profits for entrepreneurs, investors, and ordinary non-disadvantaged consumers. Specifically, they are "new ideas (products, services, and models) that simultaneously meet social needs and create new social relationships and collaborations [59], and lead to new or improved capabilities and relationships and better use of assets and resources".

The concept of social innovation has a nuanced variety of meanings, but for now, we can understand it in terms of social change. The change we refer to hides the mysterious power of collective actions, an ability that we are still far from appreciating and having the possibility of appropriating in terms of territoriality. Territory plays a relevant role in social cohesion that allows the generation of investments as determinants that transcend the geographical perspective, since the region is a set of economic and natural aspects. Political factors relate to society and nature, i.e., a socio-territorial space [60].

Contrarily, considering that every geographical space is an open and complex system, economic growth is exogenous, and social progress is endogenous. Based on this, social progress must be found in developing local potentialities, starting from exogenous processes [61], which include public policies and collective decisions. The 20th-century assumption of social innovation in the principles of economic theory is that all the functions governed for maximum utility and minimum cost exist, which excludes ecology and territorial differences. In retrospect, land, work, and capital are symbolic sources of wealth in capitalism. Simultaneously, society depends on its capacity to reproduce this production scheme from the territory.

Looking at the social economy reveals another implication of a fragmented vision of territory, environmental education, and social innovation. The region is implanted in the social dimension and, in turn, modifies the organizational structure that marks its identity. The binomial nature of territory and CEE poses the liberation and empowerment of communities by implementing consensual decisions that prioritize their interests to develop the collective action necessary to confront socio-environmental conflicts [62].

Briefly, this is what we call the paradox of social innovation, i.e., through the externalization of the processes of disruptive changes in the social order, dynamics are internalized, which leads to the rethinking of the drivers of economic development and includes the valorization of natural resources, cultural heritage, and landscapes as intangible assets of the territories. Parallelly, CEE offers the possibility of social progress supported by new forms of organization and functioning that reject the exploitation of natural resources and rescue identity and humanitarian values as components of territorial development [63,64].

## 3. Materials and Methods

This is a study conducted from hermeneutic phenomenology as a method that aims to understand the nature of the dynamics of identity in Latin American territory and its transformation through processes of social innovation that CEE supports from a socio-environmental perspective. Thus, it is fundamental to assume that regions are a dynamic interaction of resources and communities whose internal structure defines identity in social facts. Traditional dualistic opinions precede a deep understanding of perception, experimentation, and interaction with the environment throughout our existence. The relationship between human beings and nature is constructed [65].

Phenomenology from a socio-environmental perspective is not a study of environmental justice. It does not offer alternative descriptions and explanations of ecological phenomena, but rather focuses on recovering the deep meaning of the environmental concerns of Latin Americans. The phenomenological descriptions of coloniality are the interaction between territory and identity for communities from CEE. The collectivity of territory gives the character of conscience in liberalization. Here, the interaction and inter-relation front the act of emancipation implied in CEE.

Hermeneutic phenomenology, as a paradigm for comprehending identity and territory with the interaction between the community and environment, looks for Latin American realities [66]. Hermeneutic phenomenology combines phenomenology and hermeneutics to understand human experience and its interpretation, emphasizing the importance of interpretive understanding in knowledge construction [67]. Following the foundation of the choice of the hermeneutic method, the central argument of Heidegger that emphasizes the temporal and relational nature of human existence, the concept of authenticity and the interconnection of humans in their shared world, is resumed [68].

Hermeneutic phenomenology is a philosophical and methodological approach that focuses on understanding human experience and its interpretation within specific contexts, employing interpretive and reflective methodologies [69]. This approach combines the disciplines of phenomenology and hermeneutics to gain insights into the meaning and significance of human experiences [70]. Consequently, Latin American facts reveal the arrangement that underlies the sense and purpose of acts oriented toward environmental transformations [71]. Therefore, the first stage establishes theoretical notions in the Latin American context. In the second stage, the popular environmental movements are described through documents related to the experience. Finally, the third stage reflects on the selected experiences to structure the sense and meaning attributed to actions from a socio-environmental perspective.

The discussion and analysis of the meaning of the text on the subject is an integral revelation of the current interpretative errors, stripped of criticality and historicity in the face of the crisis of identity and belonging in Latin American territories that limit social innovation from a socio-environmental perspective.

*Research Methodology*

The literature review to address the research questions was conducted in five sequential stages. The study began with (1) formulating questions, (2) locating articles, (3) selecting and evaluating articles, (4) analyzing and synthesizing, and (5) presenting the results [72]. Article location and retrieval were carried out using search strings that included keywords such as environmental education, critical environmental education, social innovation, identity, territory, and Latin America. The electronic databases searched included Elsevier, ERIC, Emerald, Taylor & Francis, Springer, Wiley, EBSCO, ISI Web of Science, and Google Scholar. Additionally, articles were manually verified to meet the inclusion criteria of peer-reviewed publications from 2018 to 2022 that were available in full text online and published in English and Spanish.

The literature reviews integrated articles managed with Mendeley as a tool to avoid duplicate articles. The papers were classified, and meta-analysis techniques were used to quantitatively combine the research results addressing critical environmental education in Latin America to arrive at general conclusions. The aim was to generate results that can be used to identify key themes of critical environmental education and associated research agendas in Latin America.

Based on this information and the search criteria described in this section, a final sample of 11 articles was identified that addressed critical environmental education, social innovation, identity, territory, and Latin America. Next, the articles were imported into QSR NVivo software. Finally, QSR NVivo, along with data extraction forms, was used to conduct a thematic synthesis by coding data from complete articles from Brazil (5), Argentina (4), Mexico (2), Chile (1), and Ecuador (1).

The methodological procedure for conducting a literature review on CEE from a hermeneutic phenomenology perspective includes the following:

- Define the purpose of the review.
- Establish the overall objective of the review, such as analyzing the current state of critical environmental education in the literature.
- Formulate specific research questions to guide the review process.
  Source selection:
- Identify relevant information sources, such as academic articles, books, research reports, and documents related to critical environmental education.
- Consider including multidisciplinary sources to gain a comprehensive understanding of the topic.
  Reading and analysis of sources:
- Read and critically analyze each selected source.

- Use hermeneutic phenomenology to understand and interpret the meanings and experiences related to critical environmental education presented in the sources.
- Make annotations and summaries of key findings, highlighting relevant concepts, ideas, and perspectives.

Identification of themes and patterns:

- Organize the findings into emerging themes and subthemes.
- Look for ways and relationships among different concepts and perspectives in the literature.

Reflection and reinterpretation:

- Reflect on the identified themes and patterns in light of your research questions and the review's objectives.
- Use hermeneutics to reinterpret meanings and understand the more profound implications of the findings.

Synthesis and report writing:

- Synthesize the results of the literature review into a coherent report.
- Structure the essay logically, organizing the findings into thematic sections.
- Include an introduction that explains the context of critical environmental education and the review's objectives.
- Present the findings clearly and accurately, supported by relevant examples and citations from the reviewed literature.
- Conclude the report by summarizing the main findings and highlighting the implications and potential future research directions.

Review and feedback:

- Seek feedback and comments from colleagues or experts in critical environmental education.
- Make further revisions and modifications to the report based on the feedback.

Materials:

- Relevant literature: Gather various sources, such as academic articles, books, research reports, and documents related to critical environmental education. Ensure to include a variety of perspectives and approaches to gain a comprehensive understanding of the topic.
- Note-taking materials: Use a notebook or digital tool to take notes while reading and analyzing sources. You can use different colors or labels to organize your notes by themes, key concepts, relevant quotes, etc.
- Bibliographic management software: Utilize a tool such as Zotero, Mendeley, or EndNote to manage your bibliographic references. These programs enable you to efficiently organize and cite your sources.

Methods:

- Reflective reading: Read each selected source reflectively and critically. Pay attention to the meanings and experiences related to critical environmental education present in the texts.
- Phenomenological analysis: Apply the hermeneutic phenomenological approach to understand and interpret the meanings and experiences present in the literature. Reflect on how they relate to critical environmental education and search for patterns and connections among texts.
- Coding and categorization: As you analyze the sources, encode and categorize the emerging concepts, themes, and subthemes. Identify common elements and variations from different perspectives.
- Reflection and reinterpretation: Reflect on the findings and connections identified throughout the process. Use hermeneutics to reinterpret meanings and comprehend the more profound implications of the results in the context of critical environmental education.

- Synthesis and writing: Synthesize the results of your literature review into a coherent report. Organize the findings by themes and subthemes, providing relevant examples and citations to support your claims. Ensure to explain how the findings relate to the objectives of the review.

## 4. Results and Discussion

*Latin American Perspective on Critical Environmental Education*

Throughout history, a Latin American perspective on CEE development as a scenario of social and economic change has been an instrument of the resignification of discourses and practices based on the notion of progress, which demanded an organization and regulation of the social order to turn it into a collective benefit [11]. Therefore, development resembled the terminologies of civilization, evolution, wealth, and growth, where nations acted to generate growth and solve the effects of the Second World War [9]. This process has led to policies and economic models that have responded to capital accumulation, the concentration of power, and the domination of people [10].

CEE recognizes the conflicts and contradictions of the system. The relationships between power struggles, political systems, appropriation of nature, and desertification of life are ways of historical materiality extinction [18]. From this perspective, CEE in Latin America implies the following:

1. Recognition of the potential of the environmental wealth and biodiversity of the continent with its cultural diversity and more renewable energy matrices;
2. The fragility of democracy in Latin America; most of the nations have gone through dictatorial systems;
3. The development model adopted by governments is anthropocentric and predatory capitalism. The need to overcome the need for environmental education with a conservationist base;
4. Mining extractives are predominant in Argentina, Brazil, Bolivia, Chile, Colombia, Ecuador, Mexico, Paraguay, Peru, Uruguay, and Venezuela;
5. The traditional populations or native people suffer in defense of life;
6. The situation of poverty and, in many cases, extreme poverty threatens the daily life of the population;
7. The concentration of wealth in the hands of a few causes great economic inequality to prevail on the continent;
8. In the same nations and the Dominican Republic, Nicaragua, Venezuela, and more recently, Chile and Brazil, the population calls for democracy in the streets. This was a continental protest;
9. Crime, corruption, and drug trafficking put many lives at risk and changed the Latin American social landscape. These combined elements cause inhabitants to lose their lives. Associated with this is the omission of many governments in some cases; in others, the direct participation of alliances with the development model they assume;
10. Many migratory movements in Latin America search for employment and basic guarantees, such as health and education [14].

A study on environmental education and popular social movements in Latin America shows that the horizon of CEE in the region still requires many studies to understand reaffirmation and recognition processes in the search for an identity that assumes the diversity that characterizes the area [73]. Latin American countries continue to work from the perspective of sustainable development based on the neoliberal policies of the large agencies that determine the horizon of practices for achieving the Sustainable Development Goals (SDGs) rather than from another perspective. In Latin America, countries are guided by the logic of capitalist development models [14]. CEE is opposed to the principles and logic of neoliberal policies, as it is assumed from a worldview of social transformation.

Among the most influential macro trends in environmental education are the following: (a) a positivist epistemology guides the conservationist aspect; (b) critical dialectics oriented by critical epistemology with epistemological foundations (Marx, Frankfurt School, Freire,

and Loureiro, among others); and (c) phenomenological hermeneutics guided by the epistemologies of Gadamer, Heidegger, and many others.

The predominance of conservation-based environmental education in Latin America indicated the need for CEE as a mechanism for social innovation [74]. Thus, in this study, the epistemological position of CEE is taken as a reference, which considers that its main objective is to affirm being a social practice; as with everything that refers to human creation in history, environmental education must link ecological and social processes in the world such that it intervenes and exists in nature. Therefore, it is recognized that we relate to nature through social media, i.e., through dimensions that we create in the very dynamics of our species and that form us throughout life (culture, education, social class, institutions, family, gender, ethnicity, nationality, etc.). Thus, we are unique syntheses of relationships and a complex unity involving biological structure, symbolic creation, and the transformative action of nature [18].

From the position of Heraclitus, it is proposed that Eurocentric Western thought in its ontology was imposed as a way of being, and consequently, Latin America and Africa, among others, are seen as peripheral, colonized, and unrecognized, i.e., in the condition of not being from that point of view. One can only think of being for Parmenides, since non-being is not proposed. Specifically, they cannot be converted or transformed.

Here, we find the premises of thought in which being is affirmed as a condition for negating non-being. Contrastingly, a CEE emerged that was based on the understanding of the decolonial studies of Dussel, who defended the philosophy of liberation to recognize Latin American thought [75], which is in line with Zimmermann, who proposed assuming the role of Latin America as a condition of its emancipation [76]. Except for significant proportions, both criticize the extent to which the modern Western European projection consists of totalizing and colonizing, not only of human beings but also of knowledge present in innumerable alterities located as peripheral until then.

In the case of Latin America, this differentiation is fundamental, because it is possible to bring the categories of the struggle for the recognition of those belonging to other territories. The ethical foundation consists of a permanent affirmation: "The philosophy of liberation, by assuming the ethics of otherness as a principle, opens itself to the epiphany of the other, which is an inexhaustible mystery".

They are opening themselves to other means, assuming an attitude of listening. For Dussel, this attitude or capacity to listen to the voice of the other is called "ethical conscience" [77]. Hence, the defense in this study from the perspective of socio-environmental justice recognizes that the "non-being" is the one who is historically denied and in a situation of oppression. From a European perspective, the subjects are considered on the margins to the extent that their knowledge, philosophies, thoughts, and pedagogies are not recognized as valid. They occupy the condition of "not being" but have fought for greater participation in Western culture, economy, and politics [78].

The Latin American perspective on CEE has aimed to transform discourses and practices based on progress, questioning the prevailing anthropocentric and capitalist development model. While recognizing the environmental wealth and biodiversity of the continent, Latin America also faces challenges, such as democratic fragility, concentration of power and wealth, mining exploitation, poverty, and economic inequality. Additionally, the migration movements in Latin America reflect the search for employment and basic guarantees. CEE seeks to overcome these issues and promote sustainable development that values cultural diversity and renewable energies, involving social and popular movements in the struggle for an identity that acknowledges the region's reality and needs.

However, further studies are still needed to fully understand the reaffirmation and recognition processes in Latin America's environmental education. Governments and civil society continue to work toward a more equitable and environmentally conscious development approach, but challenges persist, including corruption, drug trafficking, and violence that threaten people's lives. On the other hand, citizen participation and demands for democracy are becoming more vital in several regional countries. In this context,

environmental education emerges as a critical tool for driving social and economic change in Latin America, recognizing the importance of environmental justice and the defense of life in all its manifestations.

In addition to the Latin American perspective on critical environmental education (CEE), it is essential to consider other approaches and methodologies used in the field. For instance, the hermeneutic phenomenology perspective and literature review are two widely employed approaches to understanding and analyzing environmental education across different contexts. Hermeneutic phenomenology allows the exploration of subjective experience and the meanings attributed to the relationship between individuals and their natural environment. On the other hand, the literature review provides a comprehensive analysis of previous research and existing theories in environmental education.

Nonetheless, these methodologies also have certain limitations. For example, hermeneutic phenomenology can be subjective and reliant on individual interpretation, which can introduce biases and challenges in generalizing the findings. Furthermore, the literature review may be constrained by the availability and quality of previous studies and the potential exclusion of alternative perspectives or approaches.

Despite these limitations, both hermeneutic phenomenology and literature reviews have significantly contributed to the field of environmental education, generating new knowledge and meaningful insights. These approaches have helped with understanding the interactions between individuals and their environment and identifying challenges and opportunities for promoting critical environmental education. Furthermore, they have highlighted the importance of considering multiple perspectives and approaches in pursuing sustainable and equitable solutions.

Ultimately, these approaches and methodologies impact a broader field by fostering critical reflection, interdisciplinary dialogue, and collaboration among different stakeholders, such as researchers, educators, policymakers, and local communities. Furthermore, by integrating new knowledge and lessons, environmental education's theoretical and practical foundations are strengthened, contributing to addressing environmental challenges and promoting sustainability in the Latin American context.

## 5. Conclusions

CEE is very fruitful for allowing us to understand Latin America from its potentialities, historical distortions, forms of colonization, modes of production, exploitation of nature, political commitments, alliances, growing inequalities, and extreme poverty. Beyond presenting the region's contradictions, CEE contributes decisively to the understanding of the territory and the identification of alternatives of social innovation and sustainability favoring community life, recognizing new ways of being and living together in Latin America. The way to face the challenges of the dynamics of the communities from their territories assigns a role to environmental education as a lens that makes visible the solutions that a more sustainable economy demands, focused on processes of social innovation with the capacity to scale to achieve environmental justice and social equality. For this reason, critical environmental education is based on comprehensive training accompanied by the sustainable use of natural resources and the implementation of more efficient and environmentally friendly production and consumption practices.

The most significant problem affecting Latin America is social inequality. CEE assumes an interpretative force that enables alternate ways of thinking about coexistence and life in the region. Thus, from the critical, reflective, and problematizing dimensions, CEE demands alternatives that confront the logic of the predatory capitalist production model, with the strengthening of identity and belonging of people and with a search for collective and cooperative horizons in the reorientation of life and the humanity–nature relationship.

Understanding the historical dynamics of the territories, the relations of human and non-human nature domination, and the demystification of neoliberal economic agendas constitutes the axis of the dependence of Latin American countries. Thus, CEE is a decolo-

nizing engine that calls for an ethical recognition and valuation of the knowledge of the people who have lived in the region for a long time.

Thus, one viable alternative is social innovation, with its potential for territorial development. This paradigm indicates the possibility of reconciling environmental and economic growth, adding cooperative perspectives to the territories, and developing innovative services based on the social needs of the territories. These are concrete possibilities for confronting social inequality, losing cultural identity, reinforcing autonomy, and constructing participative emancipatory projects. Territories are spaces where social relations take on unique contours and fruits of the differentiated cultural–historical processes that give rise to the rich socio-environmental diversity of Latin America. This socio-environmental diversity reproduces life through social, historical, political, economic, and cultural interactions. This conjunction is manifested in the development of local contexts. Therefore, any measure of growth must consider the locality to make the development concept more flexible. The product is not purely economic but imbued with humanity and nature, where justice and sustainability are unceasingly sought-after objectives.

In view, CEE plays an essential role in bringing the necessary attention to the context where people live, reflecting on the historical–cultural process and its effects and providing solutions to overcome social inequalities. Furthermore, such development is educational, i.e., it recognizes the importance of learning from critical reflection on the territory and the possibility of its local transformation in a global context.

Social innovation is the product of criticism of the experienced reality and mobilization for a search for fairer and more sustainable solutions resulting from these differentiated educational processes in the development context. Therefore, social innovation is the creative, dynamic, and pragmatic result of the learning process from CEE in the context of development.

The conceptual approach of territory, development, social innovation, and CEE are essential to instrumentalize them as drivers for overcoming the economic model that amplifies social inequalities and promotes environmental problems that are often irreversible in the short term. It can generate profound territorial, socio-environmental, cultural, and social changes.

Applying hermeneutic phenomenology and a literature review in this study has provided valuable insights into the critical field of CEE in Latin America. By exploring Latin America's potential, historical distortions, modes of production, exploitation of nature, political commitments, growing inequality, and extreme poverty, CEE has contributed to our understanding of the region and its contradictions. Furthermore, CEE has fostered an identification of alternatives for social innovation and sustainability that favors community life and recognizes new ways of being and coexisting in Latin America.

Moreover, the most significant problem affecting Latin America, social inequality, has been highlighted. CEE assumes an interpretative force that enables thinking about alternative forms of coexistence and life in the region. From critical, reflective, and problematizing dimensions, CEE demands alternatives that challenge the predatory capitalist production model's logic, strengthen individuals' identity and belonging, and seek collective and cooperative horizons in reorienting life and the relationship between humanity and nature.

In conclusion, hermeneutic phenomenology and a literature review provided a solid theoretical and methodological framework for this study, enabling a profound analysis of critical environmental education in Latin America. These approaches have revealed the region's issues, challenges, and possibilities. Furthermore, they have highlighted the importance of engaging local communities and social and popular movements while promoting a holistic approach that recognizes the interconnectedness between humanity and nature. This study offers important lessons and calls to action for the broader environmental education and sustainable development fields in Latin America and beyond.

**Author Contributions:** Conceptualization, R.F.d.S., A.D.T.-R., V.A.P., L.R.C. and M.J.B.; methodology, R.F.d.S. and A.D.T.-R.; formal analysis, R.F.d.S., A.D.T.-R., V.A.P., L.R.C. and M.J.B.; investigation, A.D.T.-R., R.F.d.S., V.A.P., L.R.C. and M.J.B.; writing—original draft preparation, R.F.d.S., A.D.T.-R.,

V.A.P. and M.J.B.; writing—review and editing, R.F.d.S. and A.D.T.-R.; supervision, R.F.d.S. and A.D.T.-R.; project administration, R.F.d.S. and A.D.T.-R. All authors have read and agreed to the published version of the manuscript.

**Funding:** This research received no external funding, but the APC was partially covered by Instituto Politécnico Nacional.

**Institutional Review Board Statement:** Not applicable.

**Informed Consent Statement:** Not applicable.

**Data Availability Statement:** Not applicable.

**Conflicts of Interest:** The authors declare no conflict of interest.

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
