# Peer review of "Critical Environmental Education in Latin America from a Socio-Environmental Perspective: Identity, Territory, and Social Innovation"

_sustainability, doi:10.3390/su15129410_

Round 1

Reviewer 1 Report

Dear authors,

Please take into consideration the following issues:

1. In general, the structure of a scientific paper comprises the following main sections: INTRODUCTION, LITERATURE REVIEW, RESEARCH METHODOLOGY, RESULTS, DISCUSSION and CONCLUSIONS. Thus, the paper should be restructured.

2. The TITLE of the paper seems appropriate.

3. It is not appropriate to use acronyms in the ABSTRACT.

4. The aim is clearly presented in the INTRODUCTION.

5. The RESEARCH METHODOLOGY section should be significantly improved. The research model and the hypotheses are missing. The authors have to better explain the research method they used and its relevance for the study. 

6. In CONCLUSIONS, the authors should better outline why the outcomes of their study are relevant from a scientific point of view. Also, they should emphasize the relationships with other studies.

Good luck!

The use of English can be improved.

Author Response

Dear reviewer,
We greatly appreciate your valuable comments, which have been helpful in restructuring this article.
My best regards,

Response to Reviewer 1 Comments

Point 1: In general, the structure of a scientific paper comprises the following main sections: INTRODUCTION, LITERATURE REVIEW, RESEARCH METHODOLOGY, RESULTS, DISCUSSION and CONCLUSIONS. Thus, the paper should be restructured.

Response 1: We commented that the sections of the article were restructured and the methodology was improved and better explained.

Point 2: The TITLE of the paper seems appropriate.

Response 2: There was no change in title.

Point 3: It is not appropriate to use acronyms in the ABSTRACT

Response 3: Acronyms have been removed from the abstract.

Point 4: The aim is clearly presented in the INTRODUCTION.

Response 4: The introduction was complemented in view of the other reviewers.

Point 5: The RESEARCH METHODOLOGY section should be significantly improved. The research model and the hypotheses are missing. The authors have to better explain the research method they used and its relevance for the study.

Response 5: The methodology section was improved, where it is explained how we carry out the work.

Point 6: In CONCLUSIONS, the authors should better outline why the outcomes of their study are relevant from a scientific point of view. Also, they should emphasize the relationships with other studies..

Response 6: This point is better described because the results are relevant and we relate them to other studies and their need.

Reviewer 2 Report

This paper addressed an important and interesting topic: Critical environmental education in Latin America from a socio-environmental perspective: identity, territory, and social innovation. Overall, this paper is not rigorous enough. The authors must do a better job in terms of: 1) what are the data or theories used? 2) What methodologies you have used? 3) What are the main contributions of this paper?

For more detailed comments, please see the attached file. 

For more detailed comments, please see the attached file. 

Author Response

Dear reviewer,
We greatly appreciate your valuable comments, which have been helpful in restructuring this article.
My best regards,

Response to Reviewer 2 Comments

Point 1:  what are the data or theories used?

Response 1:. 1) Critical Environmental Education is an approach that goes beyond traditional environmental education by addressing the social, political, and economic dimensions of environmental problems. It emphasizes the examination of power structures, social inequalities, and historical contexts that contribute to environmental problems and their disproportionate impacts on marginalized communities.

From hermeneutic phenomenology, as a philosophical and interpretive approach, the experiences and perceptions of the members of a community are understood. With this approach, the study seeks to delve into how individuals perceive and interact with nature, the territory and the processes of social innovation from a socio-environmental perspective.

The principles of Critical Environmental Education (CEE) and hermeneutical phenomenology are used to analyze the role of the CEE in Latin America, reveal historical distortions, explore the socio-environmental impacts of climate change, and identify processes of social innovation that can contribute to the community and environmental empowerment justice.

Point 2: What methodologies you have used?

Response 2: Two main methodologies are used in the article:

Bibliographic review: The study involves a bibliographic review of articles published between 2018 and 2022. Therefore, relevant articles on critical environmental education, territory, and social innovation are collected and analyzed. Knowledge from various sources is then synthesized and integrated to provide an overview of the role of critical environmental education.

The study uses hermeneutic phenomenology as a methodology to deepen the understanding of the perception and interaction with nature, the territory, and the processes of social innovation from a socio-environmental perspective. By using this methodology, the study aims to discover individuals' underlying perceptions and interpretations in relation to the socio-environmental aspects of critical environmental education in Latin America.

Point 3:  What are the main contributions of this paper?

Response 3: The main contribution of the paper lies in addressing the contribution of Critical Environmental Education in Latin America and explaining how the understanding of historical distortions, the socio-environmental impacts of climate change, and the potential of social innovation has a transformative approach that addresses environmental problems justice, the power of community identity and cultural values that promote actions within the framework of the principles of sustainable development with the conviction of creating a more equitable future from the man-nature relationship.

Comments on the PDF

Commented [A1]: Should be: Critical Environmental Education

Response A1: This point in the text was correct.

Commented [A2]: In the Introduction, you also need to explain, at least briefly: what are your main results and contributions? How the rest of the paper is organized?

Response A2: The main results and contributions were explained in the introduction, as well as how this study is organized.

Commented [A3]: How is your "literature review" organized? What is the logic behind it? Information should be provided here.

Response A3:  We explain in the text, as your suggestion, how the literature review was organized and information was given for a better understanding.

Commented [A4]: Why each paragraph here is just one long sentence?

Response A4:  The paragraphs have been corrected.

Commented [A5]:  Need to improve writing here.

Response A5:  The paragraphs have been corrected.

Commented [A6]:  This section is too short. I suggest you reorganize your paper so that much more explanations are provided here about this paper's data and methods.

Response A6:  This section has been improved and reorganized.

Commented [A7]:  These don't seem to be the results of THIS paper..

Response A7:  This section has been reorganized.

Reviewer 3 Report

This paper presented the environmental problems in the view of the justice and critical perspective. By collecting a large amount of literature and comparing different evaluation methods, this paper summarizes CEE situation in Latin America. Before considering for publication, some revisions should be made. First, the “Abstract” section should contain more information about the topic and the conclusion. Second, the “Introduction” was too simple. The importance of this research should be emphasized. And more background description should be added. Finally, the format of the references should be unified based on the guideline of “Sustainability”.

Author Response

Dear reviewer,
We greatly appreciate your valuable comments, which have been helpful in restructuring this article.
My best regards,

Response to Reviewer 3 Comments

Point 1:  This paper presented the environmental problems in the view of the justice and critical perspective. By collecting a large amount of literature and comparing different evaluation methods, this paper summarizes CEE situation in Latin America. Before considering for publication, some revisions should be made. First, the “Abstract” section should contain more information about the topic and the conclusion. Second, the “Introduction” was too simple. The importance of this research should be emphasized. And more background description should be added. Finally, the format of the references should be unified based on the guideline of “Sustainability”.

Response 1:. The summary has been improved and more information has been entered.

We commented in the introduction the importance of the work and the necessity of this study.

We follow the MDPI reference format.

Reviewer 4 Report

In general, it is an interesting paper that could be useful in some similar contexts. However, the paper requires extensive revision and the following issues should be addressed:

1.       The Abstract should communicate more information about the study. In revising the Abstract, the authors should bring forward literature gaps that this study fills, topic importance, methods and clear-cut findings. Moreover, in describing the study aim, the word “aim” would perhaps be more suitable than “objective”. The suitability of ‘reflect on’ should also be reconsidered.

2.       In general, the Introduction requires revision as it does not establish the need for performing this study. In other words, the authors should explain what drove this study and why it is important. Moreover, the novelty and contribution of the work should be pointed out.

3.       In the Introduction, it is also necessary to explain the literature gap that the study tries to fill and show how this study is different from others in the field.

4.       Subsection 2.1 of the literature review requires revision; for example, four paragraphs are way too short and seem disconnected from each other. More linking sentences should be added to help with the flow of the text. In addition, the text needs to be enriched with more literature sources in order to substantiate the provided information.

5.       I do not think that the methodology used in this study has been explained properly.  In Materials and methods, Hermeneutic Phenomenology is described quite vaguely and it is not shown how this method better fits the studied topic. In addition, the authors should explain the exact steps they followed to perform this study so that the study is replicable. To that end, the following works may be very helpful:

Heidegger M. Being and Time. Malden, MA, USA: Blackwell Publishing, 1962.

Merleau-Ponty M. Phenomenology of Perceptions. London: Routledge and Keegan Paul, 1962.

Laverty, S. M. (2003). Hermeneutic phenomenology and phenomenology: A comparison of historical and methodological considerations. International journal of qualitative methods, 2(3), 21-35.

Sloan, A., & Bowe, B. (2014). Phenomenology and hermeneutic phenomenology: The philosophy, the methodologies, and using hermeneutic phenomenology to investigate lecturers’ experiences of curriculum design. Quality & Quantity, 48, 1291-1303.

Standing, M. (2009). A new critical framework for applying hermeneutic phenomenology. Nurse Researcher, 16(4).

6.       Once the authors revise their methodology, they could also consider restructuring results and try to present them in an engaging and lively manner.

7.       After Results, a standalone Discussion section should be added in order to upgrade the quality of the paper. In this section the current study should be compared to others in the field. In addition, the authors should assess their methodology and show its limitations. Moreover, it would be highly interesting to highlight new knowledge and lessons and explain how it affects the wider field.

8.       In addition, a Conclusion section should be added where the main conclusion for theory and method development will be stated loud and clear. Readers need to understand how this study is important and its key messages.

The text needs editing by a professional editing service as there are numerous errors in terms of grammar and syntax. In addition, vocabulary is misused and many sentences do not communicate the intended meanings and need revision.

Author Response

Dear reviewer,
We greatly appreciate your valuable comments, which have been helpful in restructuring this article.
My best regards,

Response to Reviewer 4 Comments

Point 1:  The Abstract should communicate more information about the study. In revising the Abstract, the authors should bring forward literature gaps that this study fills, topic importance, methods and clear-cut findings. Moreover, in describing the study aim, the word “aim” would perhaps be more suitable than “objective”. The suitability of ‘reflect on’ should also be reconsidered.

Response 1:. The Abstract has been improved to communicate more information about the study and suggestions have been carried out in the text.

Point 2:  In general, the Introduction requires revision as it does not establish the need for performing this study. In other words, the authors should explain what drove this study and why it is important. Moreover, the novelty and contribution of the work should be pointed out.

Response 2:. The contribution of this work and the motivation for writing this study were explained.

Point 3:  In the Introduction, it is also necessary to explain the literature gap that the study tries to fill and show how this study is different from others in the field.

Response 3:. This point is explained at the end of the introduction.

Point 4:  Subsection 2.1 of the literature review requires revision; for example, four paragraphs are way too short and seem disconnected from each other. More linking sentences should be added to help with the flow of the text. In addition, the text needs to be enriched with more literature sources in order to substantiate the provided information.

Response 4:. This point has been corrected in the text.

Point 5:   I do not think that the methodology used in this study has been explained properly.  In Materials and methods, Hermeneutic Phenomenology is described quite vaguely and it is not shown how this method better fits the studied topic. In addition, the authors should explain the exact steps they followed to perform this study so that the study is replicable. To that end, the following works may be very helpful:

Heidegger M. Being and Time. Malden, MA, USA: Blackwell Publishing, 1962.

Merleau-Ponty M. Phenomenology of Perceptions. London: Routledge and Keegan Paul, 1962.

Laverty, S. M. (2003). Hermeneutic phenomenology and phenomenology: A comparison of historical and methodological considerations. International journal of qualitative methods, 2(3), 21-35.

Sloan, A., & Bowe, B. (2014). Phenomenology and hermeneutic phenomenology: The philosophy, the methodologies, and using hermeneutic phenomenology to investigate lecturers’ experiences of curriculum design. Quality & Quantity, 48, 1291-1303.

Standing, M. (2009). A new critical framework for applying hermeneutic phenomenology. Nurse Researcher, 16(4).

 Response 5:. Thank you very much for the suggestion of the authors and within point 3. Methodology, we cite these authors.

The steps of the methodology were explained according to the comment.

Point 6:  Once the authors revise their methodology, they could also consider restructuring results and try to present them in an engaging and lively manner.

Response 6:. The methodology and results section has been restructured.

Point 7:   After Results, a standalone Discussion section should be added in order to upgrade the quality of the paper. In this section the current study should be compared to others in the field. In addition, the authors should assess their methodology and show its limitations. Moreover, it would be highly interesting to highlight new knowledge and lessons and explain how it affects the wider field.

Response 7:. In the Results and Discussion section, we evaluated the methodology, showing its limitations.

We leave the results and discussion in point 4, but each topic is explained in detail.

Point 8:    In addition, a Conclusion section should be added where the main conclusion for theory and method development will be stated loud and clear. Readers need to understand how this study is important and its key messages.

Response 8:. The Conclusion section was added and this part was explained in detail.

Round 2

Reviewer 1 Report

Dear Authors,

You made the necessary improvements.

Good luck!

Minor changes are required.

Author Response

Dear Reviewer,
Thank you very much for your contribution. Your comments helped to improve our paper. The final version of this document underwent English correction.

My best regards

Reviewer 2 Report

I think the paper has made enough improvement to be published.

The authors should try again to improve the paper writing.

Author Response

(The authors gave the same response as above.)

Reviewer 3 Report

It can be accepted.

Author Response

(The authors gave the same response as above.)

Reviewer 4 Report

The authors have addressed comments comprehensively.

Some minor editing could be beneficial. In specific, the authors should ensure that they use properly vocabulary although cases of misuse are very few.

Author Response

(The authors gave the same response as above.)
